# The Molecular Mechanism of Body Axis Induction in Lampreys May Differ from That in Amphibians

**DOI:** 10.3390/ijms25042412

**Published:** 2024-02-19

**Authors:** Galina V. Ermakova, Aleksandr V. Kucheryavyy, Andrey G. Zaraisky, Andrey V. Bayramov

**Affiliations:** 1Shemyakin-Ovchinnikov Institute of Bioorganic Chemistry, Russian Academy of Sciences, Moscow 117997, Russia; gala1559@gmail.com; 2Severtsov Institute of Ecology and Evolution, Russian Academy of Sciences, Moscow 119071, Russia; scolopendra@bk.ru; 3Department of Regenerative Medicine, Pirogov Russian National Research Medical University, Moscow 117997, Russia

**Keywords:** embryonic induction, lamprey, Noggin, Chordin, axis development

## Abstract

Lamprey homologues of the classic embryonic inducer Noggin are similar in expression pattern and functional properties to Noggin homologues of jawed vertebrates. All *noggin* genes of vertebrates apparently originated from a single ancestral gene as a result of genome duplications. *nogginA*, *nogginB* and *nogginC* of lampreys, like *noggin1* and *noggin2* of gnathostomes, demonstrate the ability to induce complete secondary axes with forebrain and eye structures when overexpressed in *Xenopus laevis* embryos. According to current views, this finding indicates the ability of lamprey Noggin proteins to suppress the activity of the BMP, Nodal/Activin and Wnt/beta-catenin signaling pathways, as shown for Noggin proteins of gnathostomes. In this work, by analogy with experiments in *Xenopus* embryos, we attempted to induce secondary axes in the European river lamprey *Lampetra fluviatilis* by injecting *noggin* mRNAs into lamprey eggs in vivo. Surprisingly, unlike what occurs in amphibians, secondary axis induction in the lampreys either by *noggin* mRNAs or by *chordin* and *cerberus* mRNAs, the inductive properties of which have been described, was not observed. Only *wnt8a* mRNA demonstrated the ability to induce secondary axes in the lampreys. Such results may indicate that the mechanism of axial specification in lampreys, which represent jawless vertebrates, may differ in detail from that in the jawed clade.

## 1. Introduction

The mechanisms of early embryonic differentiation and neural induction have been studied in developmental biology since the first quarter of the 20th century. The classical model of neural induction, originating from the experiments of H. Spemann and H. Mangold conducted on amphibian embryos [1], assigns a central role in neural differentiation to the inductive center (Spemann organizer). As established later, cells of the embryonic “organizer” express antagonists of the BMP signal (proteins Noggin, Follistatin, Chordin), allowing neural differentiation and inducing the formation of secondary body axes [2,3,4,5,6,7,8]. Organizational centers have also been found in representatives of other classes of vertebrates (the embryonic shield in teleost fish, Hensen’s node in birds and mammals), but the question of the functional usefulness of these structures as an organizer remains controversial [9].

In some organisms, suppression of BMP signaling does not play a major role in neural induction. Thus, in chicks, signaling pathways associated with fibroblast growth factors FGF and, possibly, Wnt play a key role in neural induction, while the BMP pathway is described to be a modulator, mostly involved in establishing the boundaries of the neural plate [10]. In chordate relatives, i.e., hemichordates and echinoderms, neural differentiation is limitedly sensitive to changes in BMP signaling, and the Wnt pathway may play a significant role [11,12]. One of the possible explanations for the observed diversity in the properties of organizers may be the difference in the spatial organization of the embryos and their patterning in different groups of vertebrates. The molecular activities of the classical Spemann organizer in amphibians look more like a special case in this series than a basic scenario [9].

The observed variability in the mechanisms of embryonic induction in different evolutionary branches of vertebrates has increased the interest in studying these mechanisms in the extant representatives of the most basal clade that diverged from jawed vertebrates, i.e., the cyclostomes (lampreys and hagfish).

In the classical induction model, one of the key inductive factors is the *Noggin* gene family, the discovery of which by Richard Harland’s group in 1992 was a significant milestone in the history of molecular developmental biology [3]. Noggin was one of the first described proteins that was shown to induce the formation of secondary body axes when expressed ectopically in *Xenopus* embryos. Further studies showed that vertebrates have several *noggin* genes and that they are crucial regulators of cell differentiation and early development of vertebrate embryos. *Noggin*s play key roles in a wide range of developmental processes, including the development of the forebrain [13,14]. At the molecular level, the gnathostomes’ Noggin1 and Noggin2 showed the ability to not only suppress BMP but also modulate the Nodal/Activin and Wnt signaling pathways. Recently, an important role in the anti-BMP activity of the Spemann organizer was also revealed for Noggin2 [15]. In contrast to Noggin1 and Noggin2, Noggin4 contains a number of mutations that impair its ability to bind BMP and inhibit the BMP signaling pathway [16]. Consistent with this, ectopic Noggin4 cannot induce additional axes in *Xenopus* embryos [16].

Recently, we described four *noggin* genes, *nogginA*, *nogginB*, *nogginC* and *nogginD*, in lampreys and studied their physiological functions by injecting their synthetic mRNAs into *X. laevis* embryos [17]. We proposed that these four noggin genes were derived from one ancestral *noggin* gene as a result of two rounds of genomic duplications before the separation of cyclostomes and gnathostomes in the early evolution of vertebrates [17]. This scenario is supported by the fact that only one *noggin* gene is described in the closest relatives of vertebrates—lancelets and tunicates.

In terms of their expression patterns and functional properties, lamprey *noggin* proteins have demonstrated homology with gnathostome *noggin* proteins. We also confirmed the functional conservation of lamprey Noggin proteins by demonstrating their ability to induce complete secondary axes in *Xenopus* embryos [17]. According to the “two inhibitor model” for head induction, this indicates the ability of Noggins to suppress the activity of at least two signaling pathways—BMP and Wnt [18,19,20,21]. Such functional conservatism of genes between different groups of animals is not unusual; for example, it was shown previously that the cDNA of *Hydra vulgaris Noggin* can also induce axes in *X. laevis* embryos [22].

In this study, to test the conservation of the mechanism of axial induction in lampreys, we attempted to induce secondary body axes in the embryos of the European river lamprey *L. fluviatilis* by injecting synthetic *Noggin* mRNA into lamprey embryos. In general, these experiments were designed similar to ones about mRNA microinjections into *X. laevis* embryos, with the exception of a number of technical details related to the structural features of lamprey eggs (smaller size, lack of pigmentation, dense outer shell).

The unexpected lack of secondary axis induction in the lamprey experiment suggests, in our opinion, that the mechanisms of early axis development in lampreys are likely to be different from those in amphibians.

## 2. Results

To study the ability of Noggin proteins to induce the formation of additional body axes in lampreys, we injected the mRNAs of four lamprey noggin proteins, *nogginA*, *nogginB*, *nogginC* and *nogginD*, into the embryos of the European river lamprey *L. fluviatilis* and the African clawed frog *X. laevis*.

As we expected, the injections of *nogginA*, *nogginB* and *nogginC* mRNAs (15 pg per embryo) successfully induced secondary axes in *X. laevis* embryos (Figure 1A–F), which is consistent with previously obtained results [17]. In contrast, injection of the same amounts of *nogginA*, *nogginB* and *nogginC* mRNAs (10–15 pg per embryo) into lamprey embryos did not lead to the formation of additional body axes (Figure 1G,H, Table 1). Taking into account the fact that the average volume of *L. fluviatilis* eggs is approximately five times less than the *X. laevis* ones (the diameter of an *L. fluviatilis* egg is about 0.7–0.75 mm, and an *X. laevis* egg is about 1.2 mm), this amount of injected mRNA provides a fivefold excess of exogenous material compared to experiments on *X. laevis*. A similar absence of additional axes in lamprey was observed when *X. laevis noggin1* and *noggin2* mRNAs, the inductive properties of which have been well documented in the literature, were injected (Table 1, [3,14]). Attempts to increase the amount of injected mRNAs led to the death of the lamprey embryos. *NogginD* did not induce axis formation in either the lampreys or *X. laevis*, which is consistent with the previously described properties of NogginD and its *Xenopus* orthologue Noggin4 (Table 1, [16,17]).

Counting the frequency of secondary axis induction in lamprey is challenging, as river lamprey embryos at early stages (stages 19–23) lack pigmentation, which precludes visual revelation of the axes at these stages. The injected embryos often do not survive to the later stages, when the presence of additional axes can be determined beyond doubt. Therefore, to correctly consider secondary axes at the early stages, we fixed the embryos demonstrating severe abnormalities of development with low chance to survive until later stages. We stained the notochord of these embryos using the ISH method with a probe for the mRNA of the river lamprey *sonic hedgehog* (*shh*) gene. As in other vertebrate embryos, *shh* in lamprey is expressed in the notochord and in the ventral neural tube starting from the midneurula stage and later (from stage 23) also in the forebrain structures—in the hypothalamus and zona limitans intrathalamica (ZLI) (Figure 1K; [23,24]). The embryos that looked able to survive and develop were cultured until stages 25–27 and counted at this stage. So, Table 1 contains the sum of two groups of axes—the first counted at early stages by *Shh* ISH and the second counted at later stages by morphological effects.

To assess the specificity of the observed results obtained for *noggin*s, we decided to expand the range of tested inducers and injected mRNA of *Xenopus chordin*, *cerberus* and *wnt8a* into lamprey embryos. In tests with *Xenopus* embryos, the injection of 50 pg of *Chordin* mRNA per *X. laevis* results in 52% of embryos with secondary axes (n = 140). Injection of 50 pg of *Cerberus* mRNA per *X. laevis* embryo leads to 36% of embryos with secondary axes (n = 150). Injection of 30 pg of *wnt8a* mRNA per *X. laevis* embryo induces 64% of secondary axes (n = 120).

In this way we established that induction of the secondary axes with a frequency significantly (10–15 times) higher than that in control experiments was observed only when lamprey embryos were injected with *Xenopus wnt8a* gene mRNA (Table 1, Figure 1I,J,L,M, Appendix A).

Injections of each mRNA tested were carried out three to five times into independently obtained batches of eggs over three years. This is because lampreys are strictly seasonal animals, such that live embryos are available, and in vivo experiments are possible, only during the short spawning period (2–4 weeks a year).

A possible reason for the lack of induction of additional axes in the lampreys could be the degradation of the injected synthetic mRNA due to the very slow development rate in lampreys compared to the *X. laevis* frog. Thus, if in *X. laevis* the early gastrula stage, when Spemann’s organizer operates, is reached at 18 °C approximately 20 h after fertilization, the river lamprey embryos cultivated at 12 °C reach a similar stage only on the fourth day. This raises the question of whether a sufficient amount of injected mRNA and the protein translated on its matrix is retained in the embryo at this time. To test this hypothesis, we injected lamprey embryos with mRNAs encoding *Xenopus* Myc-tagged Noggin1 (*Xl-noggin1-Myc*) [14] and lamprey Flag-tagged BMPa (*Lf-BMPa-Flag*). The presence of the Myc and Flag tags allowed us to estimate the amount of Xl-Noggin1-Myc and Lf-BMPa-Flag translated from the injected templates at different stages by immunoblotting. Consequently, we revealed significant amounts of Noggin1 and BMPa proteins at the early neurula stage in the injected lamprey embryos (stage 17 according to [25]) (Figure 1N). Notably, no significant difference in the amount of Noggin1-Myc and Lf-BMPa-Flag was observed at a much later head outgrowth stage (stage 21) (Figure 1N). These results allowed us to conclude that proteins translated from mRNAs injected in the blastomeres of the cleaved lamprey embryos were still retained in the embryonic cells at least until the late neurula/head outgrowth stages. This conclusion was indirectly confirmed by the observed ability of *wnt8a* mRNA to induce second axes in the lampreys (see above).

In lampreys, BMP2/4a expression by ISH was observed relatively later, starting from the neurula stage (stage 18), and the expression of other BMP2/4 paralogues was almost absent at this stage [26]. Such a late onset of BMP expression, unless it is related to the technical limitations of the ISH method, potentially might reflect a possible difference in the role of the BMP pathway, and consequently its inhibitors, in axial induction in lampreys compared to amphibians. We checked BMP expression and phosphorylation of Smad1/5, known as downstream effectors of the BMP pathway. The literature data on the BMP expression pattern were confirmed (Figure 2A–C). Smad1/5 phosphorylation was observed in lampreys from the onset of gastrulation (Figure 2D). At this stage, the area of reduced nuclear staining for pSmad 1/5 appears above and lateral to the blastopore (Figure 2D, dotted line). At the neurula stage, the Smad phosphorylation was clearly suppressed in the neural plate region (Figure 2E,F).

We also analyzed the expression of all three *BMP2/4* genes in *L. fluviatilis* embryos by qRT-PCR, which is a more sensitive method than ISH, and observed the activation of expression at the mid–late gastrula stage (Appendix A) (stages 14–15 after [25]). This is slightly later than the early gastrula stage shown in Figure 2A (stage 13). At the same time, it should be noted that the total amount of RNA isolated from one embryo turns out to be very low before the onset of neurulation and only after the 17th stage begins to increase sharply (Appendix A). This phenomenon of dramatic increases in absolute RNA amount per embryo was also confirmed by qRT-PCR for the housekeeping genes *ef-1a* and *odc* (Appendix A). Thus, the low absolute amount of *BMP2/4* mRNA per embryo during gastrulation agrees with the results of ISH experiments, which showed that the BMP2/4 signal at the beginning of gastrulation did not exceed the background levels (Figure 2A).

## 3. Discussion

By injecting synthetic mRNA of lamprey *nogginA*, *nogginB* and *nogginC* and *X. laevis noggin1* and *noggin2* into European river lamprey *L. fluviatilis* embryos, we obtained an unexpected result: an absence of secondary axis induction. The ability of *noggin* genes, including lamprey *nogginA*, *nogginB* and *nogginC*, to induce secondary axes in amphibians has been repeatedly reported in the past and was demonstrated in the present work [3,14,17]. Moreover, we did not observe induction of secondary axes in the lamprey after injections of mRNA of such well-known inducers as *chordin* and *cerberus* [27,28,29,30,31]. Induction of secondary axes in the lampreys at a frequency significantly higher than the background level was observed only in the case of injections of *wnt8a* mRNA. This is consistent with data on Wnt inductive activity obtained for jawed vertebrates [32,33,34,35]. In summary, if the inductive activity of Wnt8a in the case of lamprey corresponds to its previously described activity in jawed vertebrates, then for other inducers tested in our experiments, an obvious difference in phenotypic effects between lamprey and amphibian embryos was observed.

Moreover, even in the case of *wnt8a*, induction of secondary axes was observed in a significantly lower percentage of cases in lamprey than in *Xenopus* (7% in lamprey in our experiments vs. 86% in *Xenopus* according to [33] and 64% in our *Xenopus* tests). This may indicate a limited competence of lamprey embryos to induce axial structures, but elucidation of the reasons for such limitations requires additional studying. Notably, our attempts to increase the amount of mRNA injected resulted in toxicity and the death of most lamprey embryos.

The frequency of embryos with double axes (“background level”), which we observed in the control experiments, generally corresponds to the data of Suzuki, 2016, who reported the spontaneous appearance of secondary axes in lamprey clutches. Although the author does not provide detailed statistics, he reported 25 embryos with spontaneous double axes, which, assuming the number of eggs per female is approximately ten thousand, allows one to estimate the frequency of spontaneous double axes as 0.2–0.3%. This estimate coincides well with the frequency of the background level of spontaneous axis formation observed in our experiments: 0.3–0.6% (Table 1). A more accurate assessment here is complicated by the fact that not all lamprey eggs are successfully fertilized, and not all that are fertilized develop normally [36].

Suppression of the BMP signaling is traditionally considered as one of the key functions of Noggins, Chordin and Cerberus and one of the key prerequisites for neural induction. Our data on Smad1/5 phosphorylation show that the core intracellular part of the axial induction mechanism at the level of suppression of Smad1/5 phosphorylation in lampreys does not differ from jawed vertebrates. As in lamprey, in *Xenopus* at the early gastrula stage, a pSmad-free zone is detected near the blastopore lip [15]. In the neural plate region at the neurula stage, phosphorylation of Smad1/5 in lamprey is also expectedly inhibited (Figure 2E,F). However, our data suggest that both the mechanism of activation of Smad1/5 phosphorylation at the gastrula stage and the mechanism of its subsequent suppression in the neural plate region at the neurula stage in lamprey may differ from the mechanism described in amphibians. For instance, this mechanism may be associated not only with the activity of BMP but also with other factors (Figure 2G). In *Xenopus*, *BMP4* expression is observed in the ventral part of the embryo starting from the gastrula stage [37]. In lamprey, neither McCauley and Bronner-Fraser (2004) nor we observed *BMP2/4a* expression in the early gastrula using ISH. The latter appears to be due to the very low absolute amount of mRNA in lamprey embryos at these stages (Appendix A).

In any case, it is important to note that our results indicate a different mechanism of axial induction in lampreys compared to the classical scheme described for *Xenopus* and many other animals. If the BMP level is too low to be the primary agent responsible for inhibiting axis induction on the ventral side of the lamprey’s early gastrula, it suggests that the mechanism of neural induction in lampreys differs from that in other model animals. Simultaneously, if a low level of BMP is still sufficient to inhibit the emergence of ectopic secondary axes but, as we have shown, injections of mRNA of known BMP inhibitors are not able to induce secondary axes, then this also indicates a difference in the mechanism of axis induction in lampreys compared to other animals.

The mechanisms of early signaling and differentiation of the embryo may be associated with the peculiarities of its structure and spatial organization. It is important here to note that the lamprey zygote, in its structure and pattern of expression of early markers, appears to be close to the ancestral state for vertebrates. Lamprey eggs are oligolecithal, with a holoblastic (complete) type of cleavage. This type of egg, in which the yolk is part of the embryo and does not form a separate yolk sac, is also characteristic of the closest relatives of vertebrates, lancelets and tunicates, which allows us to consider it a basic variant for vertebrates [38]. Moreover, this type of egg is assumed to be basic for all deuterostomes [39]. It is hypothesized that the appearance of meroblastic (incomplete) cleavage and eggs with a large supply of yolk occurred independently several times in the evolution of vertebrates and, as a rule, was associated with an increase in egg size. In holoblastic eggs, the increase in size is limited at a minimum by the difficulties of intracellular signaling with large volumes of yolk-rich cytoplasm, as well as the need to build extended membranes during complete cleavage. Large holoblastic eggs, in particular the eggs of the Puerto Rican leaf frog *Eleutherodactylus coqui*, have been studied as a possible intermediate form on the path to the appearance of meroblastic eggs. The size of the eggs of this frog reaches a diameter of 3.5 mm, with an average value for amphibian eggs of 1.5–2.5 mm. An interesting feature of this amphibian is its direct development, without a tadpole stage living in water. In *E. coqui* embryos, a “nutritional endoderm” has been described, which is a source of nutrients but is not structurally part of the developing embryo [38]. At the vegetal pole of sea lamprey embryos, an extraembryonic yolk mass has also been described, in which expression of the *soxB* and *soxF* genes was not detected [40]. That is, in their structure, lamprey embryos also occupy an intermediate position between the holoblastic zygotes of sturgeons and amphibians on the one hand and the yolk-rich meroblastic embryos of Chondrichthyes, bony fish and amniotes on the other. This “intermediate” position of lamprey eggs may be a reflection of their evolutionary antiquity, and the structure of the embryo can be considered a basic variant for the subsequent appearance of both holoblastic and meroblastic types of structures [41].

In contrast to our results on lamprey embryos, induction of complete second axes by the *noggin1* gene has been shown for the leaf frog *E. coqui* [42]. This demonstrates that the mere presence of the extraembryonic endoderm is not an obstacle to the induction of additional axes, and the reason for the lack of axial induction in lampreys appears to be something else.

A described feature of early signaling in amphibian embryos is the expression of the *vegT* gene, for which both maternal expression and expression at the blastula—gastrula stages are observed [40,43]. In lampreys and the studied bony fish (*Polypterus* and teleosts), a similar pattern is observed for *eomes* [41,44]. This indicates that amphibians, often treated as a classical model due to their contribution to our understanding of the mechanisms of early vertebrate development, may have unique features that are not relevant to other vertebrate groups. Thus, the study of regulatory pathways and signaling proteins operating at the early stages of primary embryonic induction in lampreys, as the oldest evolutionary representatives of vertebrates available for laboratory research today, is of great interest for understanding the basis of these processes in vertebrates in general.

## 4. Materials and Methods

### 4.1. Animals

All animal experiments in this study were performed in accordance with guidelines approved by the Animal Committee of the Shemyakin-Ovchinnikov Institute of Bioorganic Chemistry (Moscow, Russia) and were conducted in accordance with the Animals (Scientific Procedures) Act 1986 and the Declaration of Helsinki.

Adult mature individuals of the European river lamprey *L. fluviatilis* were caught in the Leningrad district. Developing embryos were obtained by artificial insemination in the laboratory. The eggs of mature females were placed in a container and activated in a solution of 0.1× MMR solution (at a temperature of 12 °C) for 3 min with constant stirring. After activation, the sperm of a mature male was added to the solution, and the eggs were incubated for 10 min with constant stirring. After fertilization, the eggs were washed twice with a solution of 0.1× MMR solution. Further incubation was carried out in Petri dishes (9 cm) at 12 °C. Stages were determined according to Tahara [25]. For in situ hybridization, embryos were fixed in MEMFA solution.

Injections of synthetic mRNA into the equatorial region of the embryo were carried out at the 4- to 16-blastomere stage, mostly at the 8-blastomere stage. In the absence of pigmentation in river lamprey embryos, identification of the dorsal and ventral regions at early stages is difficult; selection of successfully ventrally injected embryos was therefore carried out at later stages.

Clawed frog *X. laevis* embryos were obtained in the laboratory by artificial insemination; embryo culture and injection procedures were carried out as previously described in [14]. Synthetic mRNA was injected into the equatorial region of the embryo at the 4- to 16-blastomere stage, mostly at the 8-blastomere stage.

### 4.2. Obtaining cDNA and mRNA of Lamprey Noggins; Injections

Complete cDNAs of the river lamprey *nogginA*, *nogginB*, *nogginC* and *nogginD* genes were obtained by RT‒PCR using the primer pairs given in [17].

The resulting DNA was cloned and inserted into the pCS2 vector. Synthetic capped mRNA was synthesized by the mMessage Machine SP6 kit (Ambion, Naugatuck, CT, USA) for SP6 RNA polymerase.

The mRNAs of the clawed frog *noggin1*, *noggin2*, *cerberus*, *chordin* and *wnt8* genes were synthesized based on cDNAs obtained in previous studies [14,15].

A microinjection method was used to deliver synthetic mRNAs into embryos. Synthetic mRNA (10–100 ng of mRNA in 1 μL) mixed with the fluorescent dye FLD (fluorescein lysine dextran) was microinjected into developing clawed frog and river lamprey embryos using an Eppendorf injector (Hamburg, Germany). Injections into embryos were carried out at the stage of 4–8 blastomeres in a 4% Ficoll solution to minimize potential damage from the injection.

Assessment of Xl_Noggin1_Myc and Lf_BMPa_Flag protein expression by coimmunoprecipitation was carried out according to the methods of [14]. In one sample, 30 river lamprey embryos preinjected with synthetic mRNA were lysed.

Embryos were photographed using a Leica M205 stereomicroscope (Wetzlar, Germany).

### 4.3. In Situ Hybridization; pSmads Assay

In situ hybridization was performed on whole embryos fixed in MEMFA solution according to a previously described method [45]. After staining was completed, the embryos were fixed in MEMFA solution and, through a series of washes in a PBS solution, transferred to a solution of 75% glycerol in PBS and photographed.

cDNA of *Lf_BMP2/4a* for the ISH probe was obtained by RT-PCR with the following pairs of primers:



*Lf_BMP2/4a_full_Frw;* GCCACCATGCCGCCTGGTAGGCCGCT

*Lf_BMP2/4a_full_Rev;* CTAGCGGCACCCGCAGCCCT



Smad1/5 phosphorylation was tested using phospho-Smad1/5 (Ser463/465) Rabbit mAb, Cell Signaling Technology (Danvers, MA, USA) (41D10) and Anti-rabbit IgG, F(ab’)fragment—alkaline phosphatase antibody (produced in goat), Sigma-Aldrich (Burlington, NJ, USA) (A3937).

### 4.4. RT PCR and Total RNA Amount Analyses

RT PCR was performed as described in [17] with the following pairs of primers:



*Lf_BMP2/4a_RT_Frw1;* CACCTCAACTCCACGAACCA;

*Lf_ BMP2/4a _RT_Rev1;* CCACCTTCTCGTACTCGTCC.



*Lf_BMP2/4b_RT_Frw1;* GAGCACAGAAGGCGGAACAA;

*LC_ BMP2/4b _RT_Rev1;* CGAGGATGAGGGACATGAGC.



*Lf_BMP2/4c_RT_Frw1;* CTGAGAGCCAACGACTCCTG;

*Lf_ BMP2/4c _RT_Rev1;* CACGAGGTACTGGGATACCG.



*Lf_EF1alfa-_Frw;* AGAACGTGTCTGTCAAGGATGT;


*Lf_EF1alfa_Rev; TAGCCGGCATTGATCTGGCCA.*




*Lf_ODC_Frw;* CCGTCGGTATCATCGCCAAG;

*Lf_ODC_Rev;* CGAAGAGGATGCAGTTGAAG.



The total RNA measurement was carried out on a NanoPhotometer NP80 (IMPLEN, Munich, Germany). Total RNA samples for total RNA measurement were purified from 30 embryos of *L. fluviatilis* using an innuPREP RNA MiniKit2.0 (AnalytikJena, Jena, Germany) in three replicates.

## 5. Conclusions

Along with the presence of common features, such as the similarity of the inductive properties of Wnt8a in lamprey and *Xenopus*, significant differences were found between them in the abilities of Noggin and Chordin to induce the formation of secondary body axes, as well as of Cerberus to induce ectopic head structures. At the same time, the main intracellular part of the axial induction mechanism, based on the suppression of Smad1/5 phosphorylation, does not differ in lamprey and *Xenopus*. The observed difference in the ability to induce the second body axis for several key embryonic inducers in lamprey and *Xenopus* may indicate differences in the mechanisms of inhibition of Smad1/5 phosphorylation during such an induction in these species. A detailed study of these differences is of great interest for future research.

## Figures and Tables

**Figure 1 ijms-25-02412-f001:**
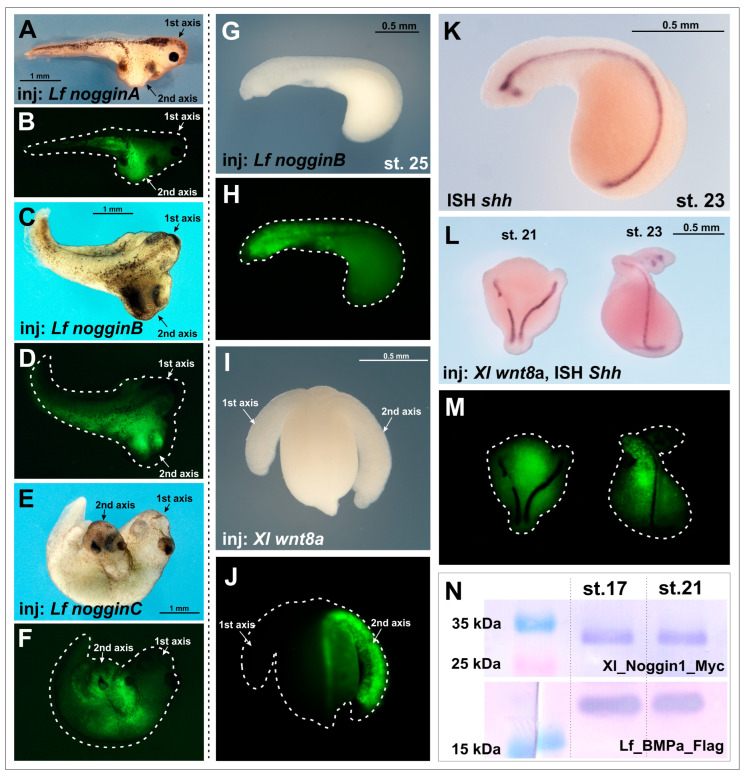
Induction of secondary body axes upon injection of *nogginA*, *nogginB*, *nogginC* and *wnt8a* mRNAs in *X. laevis* and *L. fluviatilis* embryos. (**A**–**F**) Injections of lamprey *nogginA*, *nogginB* and *nogginC* mRNAs induce secondary axes in *X. laevis*. (**G**,**H**) Injections of lamprey *nogginB* mRNA do not induce secondary axes in *L. fluviatilis* embryos. (**I**,**J**) Injections of amphibian *wnt8a* mRNA induce secondary axes in *L. fluviatilis* embryos. (**K**) *sonic hedgehog* (*shh*) expression in *L. fluviatilis* embryos detected by ISH in an uninjected control embryo. (**L**,**M**) Detection of secondary axes in *L. fluviatilis* embryos by *sonic hedgehog* (*shh*) ISH. (**N**) Proteins, translated from injected *Xl_noggin1-Myc* and *Lf_BMPa-Flag* mRNAs, are detected in lamprey embryos at stages 17 and 21.

**Figure 2 ijms-25-02412-f002:**
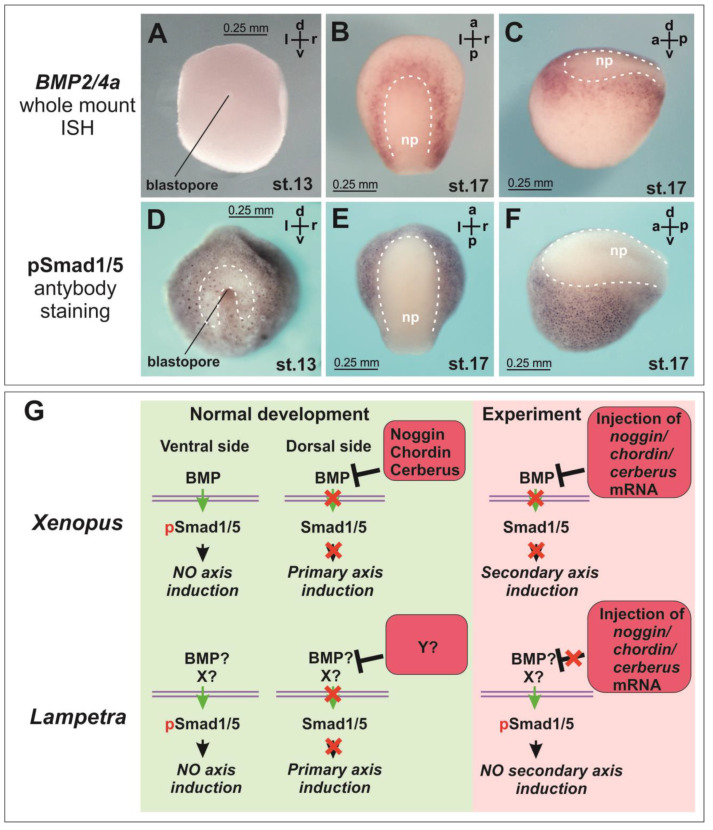
Hypothetical difference in the mechanism of BMP signaling between lamprey and amphibians. *BMP2/4a* expression by ISH (**A**–**C**) and antibody staining of Smad1/5 phosphorylation (**D**–**F**) in *L. fluviatilis* embryos. np—neural plate. (**G**) Smad1/5 phosphorylation might be modulated by factor(s) other than BMP, as injections of BMP inhibitors do not induce secondary axes in lamprey.

**Table 1 ijms-25-02412-t001:** Induction of the secondary axes upon microinjection of synthetic mRNAs of the indicated genes into European river lamprey *L. fluviatilis* embryos. A percentage approximately 10–15 times higher than that in the control experiments was observed only in the case of microinjections of *Xenopus wnt8a* mRNA. *Lf—L. fluviatilis*, *Xl—X. laevis.* Chi-square test was performed for every gene against the FLD injection control. The only significant result (*p* < 0.05) was demonstrated by *wnt8a* mRNA.

mRNA Injected	Number of Independent Batches of Eggs	Number of Injected *L. fluviatilis* Embryos	Number of Secondary Axes	Percentage of Secondary Axes Induction	*p* Value	Significance at *p* < 0.05
*Lf nogginA*	4	360	1	0.3%	0.67	-
*Lf nogginB*	4	360	2	0.6%	0.93	-
*Lf nogginC*	4	360	1	0.3%	0.67	-
*Lf nogginD*	4	360	1	0.3%	0.67	-
*Xl noggin1*	3	300	1	0.3%	0.77	-
*Xl noggin2*	5	540	2	0.4%	0.81	-
*Xl cerberus*	4	360	2	0.6%	0.93	-
*Xl chordin*	4	300	2	0.6%	0.81	-
*Xl wnt8a*	4	600	42	7%	4.16 × 10^−4^	+
FLD (injection control)	3	200	1	0.5%	-	-
Control without injections	4	3000	2	0.06%	0.05	-

## Data Availability

Data are contained within the article.

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
