# Peer review of "The Molecular Mechanism of Body Axis Induction in Lampreys May Differ from That in Amphibians"

_ijms, 2024, doi:10.3390/ijms25042412_

Round 1

Reviewer 1 Report

Comments and Suggestions for Authors

In this work, the Authors aimed to induce secondary axes in the European river lamprey Lampetra fluviatilis by injecting noggin mRNAs into lamprey eggs in vivo. The authors found along with the attendance of communal features, such as the similarity of the inductive properties of Wnt8a in lamprey and Xenopus, important changes were found between them in the abilities of Noggin and Chordin to induce the formation of secondary body axes, as well as of Cerberus to induce ectopic head structures. Furthermore, they confirmed that the main intracellular part of the axial induction mechanism, based on the suppression of Smad1/5 phosphorylation, does not differ in lamprey and Xenopus. 

The work sounds scientifically original. The discussion of the results is convincing but there are some discrepancies and corrections that need to be addressed. 

Lines 75-79: the authors “proposed that these four noggin genes were derived from one ancestral noggin gene as a result of two rounds of genomic duplications before the separation of cyclostomes and gnathostomes in the early evolution of vertebrates”. Reference or data supporting this statement are required. 

Lines 83-85: “According to the contemporary view….BMP and Wnt (Glinka et al. 1997, 1998)”. I think more recent references are required. 

Lines 96-104: in my opinion, this description of the results fits better in the discussion. 

Lines 272-274: this paragraph is not necessary in the results section. 

Lines 310-311: are there no more updated references? In general, the work needs more recent literature data. 

Author Response

Response to Reviewer 1

First of all, we would like to thank the Reviewer 1 for careful consideration of our manuscript and constructive comments. We have prepared detailed responses and a revised version of the manuscript (all changes are marked in red in the revised text file).

Reviewer 1

In this work, the Authors aimed to induce secondary axes in the European river lamprey Lampetra fluviatilis by injecting noggin mRNAs into lamprey eggs in vivo. The authors found along with the attendance of communal features, such as the similarity of the inductive properties of Wnt8a in lamprey and Xenopus, important changes were found between them in the abilities of Noggin and Chordin to induce the formation of secondary body axes, as well as of Cerberus to induce ectopic head structures. Furthermore, they confirmed that the main intracellular part of the axial induction mechanism, based on the suppression of Smad1/5 phosphorylation, does not differ in lamprey and Xenopus. 

The work sounds scientifically original. The discussion of the results is convincing but there are some discrepancies and corrections that need to be addressed. 

Lines 75-79: the authors “proposed that these four noggin genes were derived from one ancestral noggin gene as a result of two rounds of genomic duplications before the separation of cyclostomes and gnathostomes in the early evolution of vertebrates”. Reference or data supporting this statement are required. 

Our answer

In this case, we refer to our work studying the noggin genes of lampreys - Ermakova et al., 2020 https://doi.org/10.1038/s42003-020-01234-3

Reviewer 1
Lines 83-85: “According to the contemporary view….BMP and Wnt (Glinka et al. 1997, 1998)”. I think more recent references are required. 

Our answer

We have provided links to classic papers on this subject (see also the more detailed commentary below).

We have now added several references from more recent years, but they also include references to the originally sited “classic” works by Glinka et al, 1997, 1998.

Reviewer 1
Lines 96-104: in my opinion, this description of the results fits better in the discussion. 

Our answer

At the end of the Introduction, we feel it is appropriate to provide a brief summary of the paper to guide the reader (in this case, the Introduction acts as an extended and more contextualised abstract).

At the request of Reviewer1, we have removed the sentences at Lines 96-104, as there is little point in transferring them to the Discussion, where there are similar thoughts with the same References (see Lines 291-299). The Introduction now reads:

The unexpected lack of secondary axis induction in the lamprey experiment suggests, in our opinion, that the mechanisms of early axis development in lampreys are likely to be different from those in amphibians.

Reviewer 1
Lines 272-274: this paragraph is not necessary in the results section. 

Our answer

We moved this paragraph to the Discussion section (see Lines 324-340).

Reviewer 1
Lines 310-311: are there no more updated references? In general, the work needs more recent literature data. 

Our answer

We have added more recent links.

We would also like to point out that the regulatory pathways (BMP, Wnt) discussed in the article and the phenotypic effects of their modulation that we describe, relate to the basic tenets of the classical model of neural induction, the main ideas of which have been confirmed over time and have not undergone significant changes in general. Noggin and Chordin were among the first to discover the mechanisms of neural induction in the early 1990s. Early articles on the topics discussed are still relevant today, and we consider citing these classic works as a tribute to the discoverers of the discussed genes and mechanisms.

Based on the fundamental nature of the regulatory signals under consideration, we initially expected to obtain the same effects in lampreys as in Xenopus and were extremely surprised that, despite our best efforts over several years, we were never able to do so.

Reviewer 2 Report

Comments and Suggestions for Authors

The current study determined that noggin, chordin, and cerberus mRNAs were ineffective in inducing the secondary axis, however wnt8a mRNA successfully generated the secondary axis in lampreys. The manuscript clearly illustrates the significant divergence between lamprey and other vertebrate models, highlighting the evolutionary distance between them. The study anticipates the presence of uniqueness and significant observations, since we are comparing it with Xenopus or other models.

All injected mRNAs are extracellular proteins that majorly act as antagonists of BMP signaling. The authors should conduct an additional experiment to validate multiple aspects. For instance, the act of ectopic or knockdown the signals and transcription factors related to bone morphogenetic protein (BMP) signaling. It is recommended to explore the overexpression and knockdown of Gsc, including nog and chrd, from another site. This investigation may offer insights into the conserved or distant signaling pathways that are necessary for axis development in lampreys and xenopus.

Author Response

Response to Reviewer 2
First of all, we would like to thank the Reviewer 2 for careful consideration of our manuscript and constructive comments. We have prepared detailed responses and a revised version of the manuscript (all changes are marked in red in the revised text file).

Reviewer 2

The current study determined that noggin, chordin, and cerberus mRNAs were ineffective in inducing the secondary axis, however wnt8a mRNA successfully generated the secondary axis in lampreys. The manuscript clearly illustrates the significant divergence between lamprey and other vertebrate models, highlighting the evolutionary distance between them. The study anticipates the presence of uniqueness and significant observations, since we are comparing it with Xenopus or other models.

All injected mRNAs are extracellular proteins that majorly act as antagonists of BMP signaling. The authors should conduct an additional experiment to validate multiple aspects. For instance, the act of ectopic or knockdown the signals and transcription factors related to bone morphogenetic protein (BMP) signaling. It is recommended to explore the overexpression and knockdown of Gsc, including nog and chrd, from another site. This investigation may offer insights into the conserved or distant signaling pathways that are necessary for axis development in lampreys and xenopus.

Our answer

We totally agree with Reviewer 2 that stronger result would be the detailed description of the lamprey mechanism of axis induction rather just demonstration of its differences with amphibian model. But on our opinion, these are two different tasks, both in content and in scope.

In this Article (initially submitted as Brief Report), our goal is first to confidently demonstrate the observed differences in the induction of the second axis in lampreys and amphibians by known BMP-binding proteins. The presented data were systematically accumulated by us over three years of research. Our initial expectation was the possibility of inducing axes in lampreys with Noggins, by analogy with amphibians and other gnathostomes. Surprisingly, we did not observe secondary axis induction at all. Of course, having received such results, we are planning further research into the mechanisms of axial induction in lampreys. However, such work significantly goes beyond the scope of the Brief Report and involves a detailed identification of homologues of the described inductors in lamprey (including through transcriptomic sequencing), and a study of the features of their expression and functional properties. For the Noggins, we conducted similar studies earlier (Ermakova et al., 2020).

In this paper we have taken the first step and do not pretend to provide a complete description of the mechanism of axial differentiation in lampreys. That's why we've initially chosen the format of a Brief Report.

We believe that unexpected results presented in this manuscript will be of interest to researchers working with lampreys and may stimulate efforts to elucidate the details of the mechanism of axial induction in this basally divergent clade of vertebrates. Especially in the context of Special Issue “Embryonic Development and Differentiation”.

We completely agree with the need to expand the study of different aspects of mechanisms of axis induction in lampreys. At the same time, in our opinion, both the studies already carried out and the data obtained give grounds to assert the differences between the lamprey and Xenopus, that we see in side-by-side experiments and which is the main point of the presented article.

Of course, we are planning to investigate the mechanism of axial induction in lampreys in depth in the future. Given the basal position of lampreys in the vertebrate phylogeny, these studies would be important for understanding and possibly correcting the basic knowledge of the mechanism of axial induction in vertebrates.

Unfortunately, despite our best efforts, it is technically impossible to carry out further experiments at the moment, because lampreys (unlike classical laboratory animals) are strictly seasonal animals, whose living embryos are only available during the spawning period of 2-3 weeks per year. We can reiterate that the data presented in our manuscript was systematically collected over three years and is based on more than 20 years of the experimental experience with Xenopus and approximately 8 years of experience with lampreys, supported by a number of publications (Dev. Cell, Development, CommsBio, Frontiers in Cell and Deb Biol, SciRep, etc.).

Since the data obtained came as an unexpected surprise to us, we assume that they may be also interesting to other researchers in the evo-devo field and might be useful for further studies of the basic mechanisms of embryonic induction in vertebrates.

Round 2

Reviewer 2 Report

Comments and Suggestions for Authors

Good enough